# Chemical Composition and Rheological Properties of Seed Mucilages of Various Yellow- and Brown-Seeded Flax (*Linum usitatissimum* L.) Cultivars

**DOI:** 10.3390/polym14102040

**Published:** 2022-05-17

**Authors:** Yana Troshchynska, Roman Bleha, Andriy Synytsya, Jiří Štětina

**Affiliations:** 1Department of Carbohydrates and Cereals, University of Chemistry and Technology Prague, Technická 5, 166 28 Prague, Czech Republic; blehar@vscht.cz; 2Department of Dairy, Fat and Cosmetics, University of Chemistry and Technology Prague, Technická 5, 166 28 Prague, Czech Republic; stetinaj@vscht.cz

**Keywords:** flaxseed mucilage, cultivars, polysaccharides, rheology, multivariate analysis

## Abstract

When seeds sown in the soil become wet, their hulls secrete viscous matter that can retain water and thus support germination. Flaxseed mucilage (FSM) is an example of such a material and is attractive for food, cosmetic, and pharmaceutical applications due to its suitable rheological properties. FSM consists mainly of two polysaccharides, namely, arabinoxylan and rhamnogalacturonan I, and it also contains some proteins, minerals, and phenolic compounds. The genotype and the year of the flax harvest can significantly affect the composition and functional properties of FSM. In this work, FSM samples were isolated from flax seeds of different cultivars and harvest years, and their structural and rheological properties were compared using statistical methods. The samples showed significant variability in composition and rheological properties depending on the cultivar and storage time. It was found that the ratio of two polysaccharide fractions and the contribution of less-prevalent proteins are important factors determining the rheological parameters of FSM, characterizing the shear-thinning, thixotropic, and dynamic viscoelastic behavior of this material in aqueous solutions. The yield strength and the hysteresis loop were found to be associated with the contribution of the pectin fraction, which included homogalacturonan and rhamnogalacturonan I. In contrast, the shear-thinning and especially the dynamic viscoelastic properties depended on the arabinoxylan content. Proteins also affected the viscoelastic properties and maintained the elastic component of FSM in the solution. The above structural and rheological characteristics should be taken into account when considering effective applications for this material.

## 1. Introduction

Flax (*Linus usitatissimum* L., Linaceae) is a widely cultivated plant that originated in the Mediterranean and Southwest Asia regions [1]. The main reason for the popularity of flaxseed production is its oil, which has a high content of α-linolenic acid (ω-3 fatty acid), proteins (useful during meat shortage periods), and both soluble and insoluble dietary fiber, consisting mainly of polysaccharides [2]. These and other useful components of flaxseed have attracted attention, and they serve as a starting point for considering flaxseed as a potent functional food. The water-soluble dietary fiber of flaxseed, named flaxseed gum or mucilage (FSM), comes from the secondary wall material in the outermost layer.

To protect the seed under very hot and dry conditions, a mucilaginous environment is formed around a germ to absorb water for providing seed moisture [3]. FSM is a mixture of water-soluble hydrocolloids, consisting of mainly polysaccharides (50–80%) but also some proteins (4–20%) [4]. Two polysaccharide fractions of FSM were identified: neutral (weakly acidic) arabinoxylan (AX) and acidic rhamnogalacturonan type I (RG-I) [5,6,7,8]. Xylose (68.2%) and arabinose (20.2%) are the main sugars of the neutral fraction; galactose (7.9%) and glucose (3.7%) are also included in smaller percentages [9]. Uronic acids (38.7%), rhamnose (38.3%), galactose (35.2%), and fucose (14.7%) represent the acidic fraction [10]. The highly branched flaxseed AX consists of a (1–4)-*β*-d-xylan backbone substituted with l-arabinosyl, d-galactosyl, and l-galactosyl side chains at the O-2 and O-3 positions of some backbone xyloses. Terminal residues of glucuronic acid were also found in the side chains [11,12,13]. Branched flaxseed RG-I has a backbone with an alternating sequence of 1,4-bound *α*-d-galacturonic acid and 1,2-bound *α*-l-rhamnose units. Some rhamnoses are substituted at the O-3 position with linear *α*-l-arabinan, *β*-d-galactan, and type I arabinogalactan (AG-I) [12].

FSM has unique rheological properties: it behaves as a hydrocolloid with good water retention, is capable of forming viscous aqueous solutions [14], exhibits gelling at low temperatures, and can form a thermoreversible cold-setting gel [15]. Many authors have reported that FSM exhibits pseudoplastic behavior throughout the entire shear rate range with decreasing viscosity [14,16,17,18,19]. However, at the same concentrations, FSM shows significantly lower viscosity than guar gum or locust bean gum [17,18]. It was also noted that the polysaccharide fractions exhibit various functional properties that contribute to the rheology of raw FSM. Hence, it was found that the neutral fraction exhibits shear-thinning behavior and thixotropic properties, while the acidic component shows rather Newtonian flow properties, better surface activity, and emulsifying stability [10,12,16]. Chornick [19] found a strong positive correlation between the Xyl content and apparent viscosity of FSM (*r_xy_* = 0.84). Therefore, it was concluded that the neutral fraction contributed most to the apparent viscosity.

Due to the above-mentioned physical properties, gels and films based on flaxseed mucilage can be used for various food and medical purposes [20,21,22]. In the food industry, FSM could be used as a coating agent for cheese, fruits, and vegetables [23,24,25], a fat replacement in baking [26], or a texture modifier for dairy products (cheeses, yogurt, ice creams) [27,28,29], and could also be a structure-forming agent for cereals [30,31,32]. This mucilage is also capable of stabilizing food colloid systems, for example protein suspensions (yogurts), emulsions (ice cream), and oil-in-water emulsions (salad dressing) [33,34]. Edible films and various biocomposite materials based on FSM alone or with other polysaccharides (chitosan) have demonstrated suitable viscoelastic properties [22,35,36] and selective prebiotic [37] and antimicrobial effects [21,38,39]. FSM extracts and gels are used in the cosmetics and beauty industry mainly as texturizing agents [40]. Furthermore, FSM is widely used in pharmacology and medicine due to its various biological activities, including stimulation of the immune response [41], together with its anti-obesity [42,43], anti-inflammatory [44], and suitable physical properties, including biocompatibility, biodegradability, and nontoxicity. Thus, FSM can be incorporated as a direct compression polymer, binder, or disintegrant, or a suspending, matrixing, or emulsifying agent in drug delivery systems [45,46,47]. FSM can also be used as an effective mucoadhesive carrier for nasal and buccal drug delivery [48,49,50]. Finally, FSM is capable of promoting hemostatic and wound healing functions, and could be used to build materials for bleeding and wound treatment [20,51,52].

The specificity of seed mucilage is not clear for current flax cultivars without careful experimentation and analysis because this crop has been bred to change the fatty acid composition of the seed oil, the target product, but not to affect the mucilage. Furthermore, the correlation between the variety and the properties of FSM has not been sufficiently studied, and only a few reports are devoted to this topic [14,18,53]. However, although the composition of FSM and its rheological and functional properties should be largely determined by the genotype and the variety of the flaxseed, they can also change depending on the location, weather, and growing conditions [14]. The behavior of FSM in aqueous systems depends on the composition of polysaccharides and could be viscous, elastic, or viscoelastic, depending on the amounts of neutral or acid components in different cultivars [53]. The presence of proteins, polyphenols, and minerals can also significantly affect the properties of FSM. Kaewmanee et al. [18] confirmed the connection between high viscosity and the chemical compositions of different cultivars, finding that the cultivar with the most neutral sugar content had the highest viscosity. Liu et al. [54] characterized the properties of Canadian flaxseed gums and showed that the apparent viscosity ranged from 0.048 to 2.984 Pa·s among the cultivars and depended on the monosaccharide composition. Knowing how different flax cultivars could affect the monosaccharide composition and rheological properties of FSM could lead to improving and broadening the applications of FSM in the food industry and in pharmacology. For example, the use of FSM as a thickener or gelling agent is based on its ability to improve viscosity and texture, which in turn improves the organoleptic properties of food products [55]. Prerequisites for these applications are the high apparent viscosity and dynamic viscoelasticity of this hydrocolloid.

Thus, the aim of this study was to obtain mucilages from ten samples from four flaxseed cultivars from different harvesting years to compare the compositions and rheological properties of these products using statistical methods. Based on the results obtained, the individual flax cultivars used in this work were considered as a raw material suitable for the production of FSM with certain rheological properties, which could be used for specific applications such as hydrocolloids for food and other industries.

## 2. Materials and Methods

### 2.1. Materials

In this study, ten samples of various oily yellow- and brown-seeded flaxseed cultivars were used, from the harvesting years 2015–2018, supplied by the local producer Agritec, s. r. o (Šumperk, Czech Republic) (Table 1). The cultivars also differed in their fatty acid composition, i.e., the ratio of linoleic and linolenic acids, which is an important characteristic of flaxseed oil [56,57]. Amon (2007) and Raciol (2011) are yellow-seeded cultivars developed by Agritec (Šumperk, Czech Republic). Amon is characterized by a low content of α-linolenic acid (~3–4%), but the content of linoleic acid is high (~70%). The contents of linoleic acid (~39–45%) and α-linolenic acid (~29–33%) in Raciol are medium. The brown-seeded cultivars Recital (2004) and Libra (2013), bred by Laboulet Semences (Airaines, France) and Limagrain Advanta Nederland, B.V. (Rilland, Holland), respectively, have a high content of α-linolenic acid (~54–60%) but a low content of linoleic acid (~16%). In 2019–2020, mucilages were isolated from whole seeds of the above samples and laboratory tests were carried out during the same period. 

### 2.2. Preparation of Flaxseed Mucilage

FSM was extracted from whole seeds according to the procedure described previously [40,58]. The aqueous solution was obtained by suspending flaxseeds in distilled water in a solid/water ratio of 360 g/2000 g, at a temperature of 85 °C for 30 min with continuous manual stirring using a Stephan CombiCut TC300/SK400 (Stephan, Hameln, Germany) for thermostating. The viscous mixture was filtered through a sieve to remove the seeds and some impurities. The concentration of mucilage in the filtrates was obtained as dry matter determined by a gravimetric method according to ISO 5534:2004 (Cheese and processed cheese—Determination of the total solids content) in duplicate to achieve a representative value. The extraction yield (% *w*/*w*) was expressed as the ratio between the dry matter content of the mucilage recovered and the mass of the seeds. The pH of fresh FSM extracts was measured using a pH meter with a combined glass electrode (GryF HB, Havlíčkův Brod, Czech Republic) at 22 ± 2 °C. Each sample was measured at least twice. The electrolytic conductivity of the extracts was recorded using a WTW pH/cond340 m (Xylem Analytics Germany Sales GmbH & Co. KG, Weilheim in Oberbayern, Germany).

Fresh FSM extracts were pre-frozen at −70 °C and then freeze dried in a lyophilizer Freeze Dryer ALPHA 1–4 LSC (Martin Christ Gefriertrocknungsanlagen, Osterode am Harz, Germany) for 48 hrs. During the first 24 h, the main drying procedure was performed with a shelf temperature of −65 °C and 0.470 mb vacuum, with a gradual decrease in temperature. The final drying was carried out when the indicator connected to the samples showed an increase in temperature to positive values (next 24 h). The FSM yield (% *w*/*w*) was determined as the relative weight of the dried product compared to the original weight of flax seeds.

### 2.3. Analytical Methods

Samples of freeze-dried FSM were analyzed for their carbon, hydrogen, nitrogen, and sulfur contents using an Elementar vario EL Cube (Elementar Analysensysteme GmbH, Langenselbold, Germany). Powder samples (~1–5 mg) of FSM were combusted in an oxygen stream at high temperatures up to 1200 °C. Gaseous combusted products, including nitrogen, carbon dioxide, water, and sulfur dioxide were cleaned, separated, and analyzed using a TCD detector. The accuracy of this method is determined by the manufacturer for the simultaneous analysis of 5 mg of 4-amino-benzenesulfonic acid in the CHNS module, with < 0.1% absorbance for each element. Measurements were made in duplicate.

The ash content (% *w*/*w*) of the mucilage was determined according to the AOAC method (1990). The crucible and lid were first placed in an oven at 550 °C overnight to burn out the impurities and then cooled in a desiccator for 30 min and weighed. A sample (5 g) was weighed into a crucible and heated on a low Bunsen flame with the lid half closed. When the smoke stopped, the uncovered crucible and lid were placed in the oven and heated at 550 °C overnight. The crucible was then covered with a lid to prevent the fluffy ash from falling out and cooled in a desiccator. If the sample had turned grey, the ash was weighed with the crucible and a lid; otherwise, the procedure was repeated. Measurements were made in triplicate. Mineral elements (sodium, potassium, magnesium, and calcium) were determined in ash dissolved in 0.1% nitric acid (100 mL) by atomic absorption spectrometry (AAS) with flame atomization, on an AGILENT 280 FS AA spectrometer (Agilent Technologies, Santa Clara, CA, USA).

The mucilage protein content was determined using the Kjeldahl method (AOAS, 2000). In this method, all organic nitrogen is converted to ammonium sulfate by digestion in concentrated sulfuric acid. A sample (0.5–1.0 g) was placed in a decomposition flask and Kjeldahl catalyst (5 g), a mixture of potassium sulfate and copper sulfate (9:1 *w*/*w*), and sulfuric acid (200 mL) were added. A mixture of the above chemicals without the sample was used as a control. The flask was placed in an inclined position and heated gently until foaming ceased, then quickly boiled until the solution was clear. The mixture was then cooled, and 60 mL of sodium hydroxide solution (≥16 g in distilled water) was carefully added. Under alkaline conditions, ammonia is formed, which was distilled into a solution of boric acid. The flask was connected to a digestion bulb in a condenser. The condenser tip was immersed in the receiver containing 4% boric acid and 5 to 7 drops of the indicator, and a mixture of 0.1% methyl red and 0.2% bromocresol green (1:2 *v*/*v*) dissolved in 95% ethanol. The flask was thoroughly rotated and then heated until all the ammonia was distilled off. The receiver was removed, the end of the condenser was washed, and the resulting borate anions were titrated with 0.2 mol L^−1^ hydrochloric acid. The protein content was calculated from the total nitrogen content multiplied by the general protein factor (*N*·6.25). Measurements were made in triplicate.

Neutral sugars were determined after total hydrolysis and analyzed as alditol acetates by gas chromatography with a flame ionization detector (GC-FID) [59]. Samples (1–2 mg) were hydrolyzed in 72% sulfuric acid at ambient temperature for 3 h with periodic stirring. All samples were hydrolyzed in duplicate. Distilled water (2.2 mL) was added, and hydrolysis was continued for another 2.5 h at 100 °C and then ended by cooling in an ice bath. The internal standard of 2-deoxy-d-glucose (1 mg mL^−1^) was added to the remaining solution. A volume of 1 mL was transferred and neutralized with 25% ammonium hydroxide. This solution was reduced with 15% sodium borohydride in 3 mol L^−1^ ammonium hydroxide at 30 °C for 1 h. Then, 2 × 50 μL of glacial acetic acid was added after cooling in an ice bath. Subsequently, in an ice bath, 450 μL of 1-methylimidazole and 3 mL of acetic anhydride were added to the sample, and acetylation was carried out at 30 °C for 30 min. The solution was returned to the ice bath, and 3 mL of distilled water and 2.5 mL of dichloromethane were added. For efficient extraction of alditol acetates, the solution was vigorously stirred and centrifuged, and the aqueous phase was removed under vacuum. Then, 3 mL of distilled water and 2.5 mL of dichloromethane were added, and the solution was again stirred vigorously, centrifuged, and the aqueous phase removed in a vacuum, as previously described. The resulting solution of alditol acetates in dichloromethane was washed twice with distilled water, stirred, and centrifuged, and the aqueous phase was removed under vacuum. Dichloromethane was evaporated in a Termovap TV10+ evaporative concentrator (Chromservis, Prague, Czech Republic). Alditol acetates were washed twice with 1 mL of anhydrous acetone, then evaporated and stored in dry conditions. For the analysis, these alditol acetates were dissolved in 50 L of anhydrous acetone and then injected into a Shimadzu GC 2010 GC-FID (Shimadzu, Japan) capillary column DB-225 (length 30 m, inner diameter 0.25 mm, film thickness 0.15 μm). The injector and detector temperatures were 220 and 230 °C, respectively. The oven temperature program was as follows: 200 °C for 1 min, then increasing to 220 °C (40 °C min^−1^), remaining at 220 °C for 7 min, then increasing to 230 °C (20 °C min^−1^), remaining at the final temperature of 230 °C for 1 min, for a total time of 9 min.

Uronic acids (UA) were quantified by a modified colorimetric method with *m*-hydroxybiphenyl [60]. A total of 1.2 mL of sulfuric acid containing sodium tetraborate (0.0125 mol L^−1^) was added to 0.2 mL of a sample solution containing 0.5 to 20 μg of UA. The tubes were cooled in an ice–water bath. The mixture was shaken in a vortex mixer and the tubes were heated in a boiling water bath for 5 min. After cooling in an ice–water bath, 20 μL of 0.15% *m*-hydroxybiphenyl dissolved in 0.5% sodium hydroxide was added. The tubes were shaken, and the absorbance was measured at 520 nm on a Specord 50 Plus UV–Vis spectrophotometer (Analytic Jena, Jena, Germany). Since carbohydrates interfere with the analysis, the blank probe was processed without the addition of the *m*-hydroxybiphenyl reagent, which was replaced with the same volume of 0.5% NaOH. The absorbance of the blank was subtracted from the total absorbance. Standard aqueous solutions of d-galacturonic acid were used for calibration.

### 2.4. Rheometry

The steady shear and viscoelastic properties of the lyophilized FSM samples dissolved in deionized water (1% *w*/*w*) were examined using a Kinexus rheometer (Malvern Instruments Ltd., Malvern, UK) with a double gap coaxial cylinder geometry (diameter 40 mm), at a fixed temperature of 25 °C. The dependence of the shear stress (*τ*, Pa) on the shear rate ($\dot{\gamma }$, s^−1^), as well as the time dependence (an increase in shear rate from 0.1 to 100 s^−1^ for 180 s, a constant shear rate of 100 s^−1^ for 150 s, and a decrease in shear rate from 100 to 0.1 s^−1^ for 180 s) was investigated using controlled rate mode. For dynamic rheological experiments, small-amplitude oscillatory shear rheology was carried out in the control deformation mode; the amplitude of shear deformation was 0.001. The storage modulus (*G*′), loss modulus (*G*″), and phase shift angle (*δ*) were determined over the angular frequency *ω* range from 0.1 to 10 Hz (converted to radians per second for presentation). The rheological parameters were evaluated using rSpace software (Malvern Instruments Ltd., Malvern, UK), Microcal Origin (OriginLab, Northampton, MA, USA), and Microsoft Excel (Microsoft, Albuquerque, NM, USA). All experiments were carried out in triplicate.

The forward and backward steady shear rheograms were analyzed using a power model [61] (Appendix A, Equations (A1) and (A2)). Similarly, for FSM solutions, oscillatory rheograms were also described by a power function [62,63,64] (Appendix A, Equations (A3) and (A4)). The angular frequency corresponding to the crossover of *G*′ and *G*″ moduli (*ω*_c_) is characteristic of the viscoelastic behavior of hydrocolloid solutions and may indicate the prevalence of elastic or viscous behavior or the reaching of the gel–sol transition [65,66,67]. The lower the *ω*_c_ value, the greater the elastic contribution.

Thixotropy is the ability of the viscoelastic system to return to its initial state. This characteristic was assessed by monitoring the recovery process after shear [68,69]. The thixotropic character of the FSM solutions was determined by comparing forward and backward rheograms and was described through specific parameters (see Appendix A, Equations (A5) and (A6)).

### 2.5. Statistical Methods

The data obtained on repetition are represented as mean values with standard deviations. The results were compared among the samples using the analysis of variance (ANOVA) technique, using Statistica 12.0 (StatSoft, Tulsa, OK, USA). The ANOVA indicated differences among the means, and a Tukey analysis (HSD) of the differences was used for comparisons between individual FSM samples. The bivariate correlation between the analytical values obtained for the FSM samples was also evaluated (see Appendix B). Normalized analytical data were used for multivariate statistical evaluation. Hierarchy cluster analysis (HCA; Ward method of clustering, square Euclidean distances) and principal component analysis (PCA; covariation matrix) of the normalized data were carried out. All statistical analyses were performed using Statistica 12.0 software (Statsoft, Tulsa, OK, USA). The graphical outputs, i.e., dendrograms of similarity and component score graphs, were created using Origin 6.0 software (OriginLab, Northampton, MA, USA).

## 3. Results and Discussion

### 3.1. Yield and Properties of FSM Extracts

The extraction yield of FSM obtained from different flax cultivars varied from 6.4% *w*/*w* (**lb17**) to 10.0% *w*/*w* (**a16** and **rt15**). These values followed the reports that described the extraction of FSM at 80–90 °C [7,16,58,70] and up to 100 °C [58] and showed no dependence on the cultivar (Table 2). Heating up to 80–90 °C usually leads to an increase in the extraction yield of FSM [71,72]. However, this increase in FSM yield may be due to the co-extraction of proteins and possibly some polysaccharides from deeper layers of the seed shells and endosperm. Based on pH values (Table 2), the acidity of mucilages shows notable differences. The mucilages of the brown-seeded cultivars Libra Bio and Recital were more acidic (pH 5.4–5.5) than those of the yellow-seeded cultivars Amon and Raciol (pH 5.8–6.1). However, even less-acidic FSM is mentioned in the literature for various flax cultivars (pH 6.25–6.79) [18]. The electrolytic conductivity (κ) of the FSM extracts (Table 2) extended from 1.395 mS cm^−1^ (**rt15**) to 2.138 mS cm^−1^ (**a16**). Kaewmaneea et al. [18] showed a difference between the FSM conductivities of different cultivars, but the numbers were somewhat lower compared to those of the present study (0.10–0.19 mS cm^−1^). Furthermore, Wang et al. [73] reported a conductivity of 0.165 mS cm^−1^ in untreated FSM samples. Later, Wang et al. [62] obtained a markedly higher conductivity (0.769 mS cm^−1^) in ethanol-precipitated FSM samples. It seems that conductivity depends on the cultivar, the extraction temperature, and the drying/precipitation method.

### 3.2. Composition of FSM

The composition of freeze-dried FSM isolated from flax seeds of different varieties showed significant variability. The individual components of these products are presented and discussed below.

#### 3.2.1. Ash and Elemental Composition

The ash contents of the FSM samples are summarized in Table 3. In the case of Libra Bio, the ash content was significantly higher (~13.0–15.2%) than in the other samples (~10.1–11.1%), except for **a15**, which showed the maximum ash content (~19.6%). The results obtained were higher than the expected, previously published values (3.3–8.4% or 4.80–7.23%) [14,17]. The amount of ash in FSM has previously been reported to increase slightly with increasing extraction temperature (11.5% at 25 °C and 12% at 100 °C) [40] and protein content [74]. The differences in ash content found in the current work could be influenced by climatic conditions, crop age, and type.

The contents of organic and mineral elements in the isolated FSM are summarized in Table 2. As expected, high amounts of carbon (~37.2–38.3%) confirmed the predominance of organic compounds (polysaccharides), with nitrogen (~1–3%) and sulfur (0.12–0.33%) originating mainly from proteins, and potassium (~2.8–5.9%) predominating among the mineral elements in the FSM. The mucilage of the **a15** cultivar contained the highest amount of sodium (1.52%) and the lowest amount of potassium (2.80%), and therefore a high sodium level could be responsible for this cultivar having the highest amount of ash (19.6%). Magnesium (0.17–0.47%) and calcium (0.52–0.84%) were also found, but the association of these elements with pectic compounds, i.e., homogalacturonan (HG) and RG-I, present in FSM, was not confirmed (see below). The analysis of ash and elements is useful and informative for further characterization of the FSM.

#### 3.2.2. Proteins

With the extraction temperature of 85 °C presented in this study, the protein content in the FSM samples was in the range of ~4.8–17.0% (Table 3), and there were higher amounts of proteins in the case of brown-seeded cultivars (~12.6–17.0%) in comparison with yellow-seeded cultivars (~4.8–10.4%). These results were close to those expected, and other authors have reported similar protein content in FSM [16,17,40,70,75]. It is known that extraction at higher temperatures contributes to the increase in the protein amount in a mucilaginous solution [40]. Additionally, protein content has been reported to vary significantly among flaxseed cultivars [17,18]. The proteins of FSM originate from the endosperm and are not chemically connected to polysaccharides such as AX, and this is confirmed by the weak correlation between the contents of proteins and polysaccharides in this material [19,75,76]. At the same time, proteins can be associated with polysaccharides, for example, pectins (HG, RG-I), which also enter the mucilage from the deepest layers of the seed. The Kjeldahl method characterizes the content of total proteins in the FSM well, and in contrast to common photometric methods it does not interfere with the polysaccharides present in these samples.

#### 3.2.3. Monosaccharide Composition

The composition of neutral sugars of FSM polysaccharides is summarized in Table 4. Other authors have described a similar FSM composition [7,8,14,19,40,71,72,76]. The content of neutral monosaccharides obtained by GC-FID analysis of the hydrolysates made it possible to determine the ratio of the main polysaccharides of FSM. As expected, the major sugars were xylose (Xyl) and arabinose (Ara), which are the main units of AX. Two less-prevalent sugars, rhamnose (Rha) and fucose (Fuc), were associated with pectic-like polysaccharides, mainly RG-I [9,10]. Galactose (Gal) may originate from both fractions [11,12,13], and glucose (Glc) has an unclear origin, possibly originating at least partially from starch or as a monosaccharide. The contribution of individual monosaccharides was specific to the cultivars. The FSM of the Raciol group had the highest Xyl content (~35–39 mol%), followed by Recital (~32%) and the Amon group (~25–32 mol%). The cultivar with the lowest Xyl content was Libra Bio, with a range of between 23.3 and 23.9 mol%. The Ara content correlated with the Xyl content (both are from AX), and we obtained the same sequence for this monosaccharide, i.e., Raciol (~15.4–15.7 mol%), followed by Recital (~14.8 mol%), Amon (11.5–13.3 mol%), and Libra Bio (~8.3–9.2 mol%). Therefore, the FSM of the Raciol cultivars contained more AX in comparison to the FSM of the other cultivars, and this polysaccharide was less prevalent in the FSM of Libra Bio.

The Rha/Xyl, Gal/Xyl, and Fuc/Xyl ratios are characteristic of the relative contribution of RG-I, while the Ara/Xyl ratio indicates branching of AX, because Xyl is in the backbone and Ara is in the side chains. The Rha/Xyl ratio decreased in the order of raw Libra Bio (0.35–0.54), followed by Amon (0.28–0.31), Recital (0.22), and the Raciol group (0.17–0.18). A similar relationship was found for the Gal/Xyl and Fuc/Xyl ratios. The Ara/Xyl ratio showed that the most branched AX was from Recital (0.47), with somewhat less branching in AX in the Amon and Raciol cultivars (0.39–0.45), and even less in Libra Bio (0.35–0.38). Thus, these differences in the monosaccharide composition of FSM extracted from various cultivars clearly confirmed the effect of genotype on this material. Differences in Rha/Xyl and Ara/Xyl ratios between flax cultivars were defined as markers of the contribution of RGI compared to AX [75], and these ratios varied from 0.3 to 2.2 and from 0.1 to 0.9, respectively, between cultivars. Naran et al. [12] demonstrated an Ara/Xyl ratio of 0.24. Kaewmanee et al. [18] expressed the ratio between the neutral and acid fractions based on the contents of Xyl and GalA, respectively, and showed that this ratio ranged from 0.8 to 1.13 in seven Italian flax cultivars.

According to the literature [36,40,71], the sugar composition of FSM is affected by the extraction temperature. Alix et al. [36] showed that extraction at low temperatures led to the production of more acidic polysaccharides. Additionally, the Ara/Xyl and Gal/Xyl ratios increased for extraction at high temperatures from 0.18 to 0.35 and from 0.75 to 3, respectively. These results are comparable to the results of the present study (extraction at 85 °C), but the Ara/Xyl ratio was higher and varied between cultivars from 0.35 to 0.47. Vieira et al. [72] showed that the total amount of sugars decreased with increasing temperature due to the co-extraction of proteins and other polysaccharides (starch, HG).

The content of uronic acids (UA) in FSM varied considerably from 6.88% *w*/*w* (**lb18**) to 23.85% *w*/*w* (**a16**), and decreased in the order raw Amon (13.37–23.85% *w*/*w*), Raciol/Raciol Bio (9.51–21.96% *w*/*w*), Libra Bio/Recital (6.88–21.77% *w*/*w*), but the difference between varieties was negligible due to the large overlap of these ranges. The long-term storage of flax seeds may cause a higher amount of UA in FSM due to subsequent destruction of the cell wall and the release of pectins, mainly the HG part, from the hull. The amount of UA in FSM decreased with increasing extraction temperature, which could be explained by the co-extraction of proteins [40,72]. In addition, dialyzed FSM contained significantly more UA due to the removal of small molecules [72], so that the FSMs contained UA exclusively in polysaccharides. Considering that in the present study, FSM was extracted at a high temperature (85 °C) and the products were not dialyzed, the resulting UA contents were in the range expected from earlier reports [40,72]. The photometric determination of UA is thus well applicable to FSM, since it characterizes the presence of the acidic polysaccharides HG and RG-I.

### 3.3. Rheological Properties

The forward and backward rheograms (Figure 1a,b) illustrate the dependences of shear stress (*τ*) on shear rate ($\dot{\gamma }$) for 1% aqueous FSM solutions from different flaxseed cultivars over the range of 0.1–100 s^−1^. Both these curves decreased in the order Amon, Raciol/Recital, and Libra Bio. Additionally, the flow patterns represented in Figure 1a,b show that all FSM solutions exhibit thixotropic properties, because the backward curve is located below the forward curve and characteristic hysteresis loops are formed. It is evident from the curves that this loop also decreased in the order raw Amon, Raciol/Recital, and Libra Bio. The measure of hysteresis is expressed in the values of *A*_hyst_ and *I*_hyst_ (Appendix A, Table 5), and these values were high for Amon (~108–137 Pa·s^−1^ and ~30–36%), low or intermediate for Raciol and Recital (~10–28 Pa·s^−1^ and ~6–16%), and low for Libra Bio (~3–5 Pa·s^−1^ and ~6–10%). It has been previously reported [77] that for 1% aqueous solutions of FSM previously homogenized under high pressure, the area of the hysteresis loop was significantly reduced compared to that of the original FSM solution at the same concentration, and it was not restored by heating. Consequently, the thixotropic properties of FSM can be irreversibly destroyed by the same mechanism as that mentioned above for a decrease in shear thinning.

The dependences of the apparent viscosity (*η*) on $\dot{\gamma }$ and *τ* in the forward direction are shown in Figure 2a,b. The value of *η* rapidly increased at very low $\dot{\gamma }$ (~0.1–0.3 s^−1^) up to a maximum, corresponding to the yield stress (*τ_Y_*), then slowly decreased and finally became constant (Figure 2a). The value of *τ_Y_* was estimated from the maxima in the forward curves of the dependence of *η* on *τ* (Figure 2b); the results are shown in Table 5. The yield stress obtained from these plots was maximal for the Amon cultivar (~268–571 mPa), intermediate for Raciol/Recital (~35–153 mPa), and minimal for Libra Bio (~3–5 mPa). Such striking differences can be explained by the features of the FSM of these cultivars, namely, the composition and structure of macromolecules, primarily polysaccharides, and the interaction between them leading to the formation of supramolecular complexes and aggregates. For all samples, and especially for the Amon cultivar, the apparent shear-induced thickening that occurred at low shear rates (elastic region) on the forward flow curve can be explained by the structural strengthening of polysaccharide networks due to the orientation and stretching of chains, thereby increasing the number of intermolecular contacts [20,78,79]. In contrast, at higher shear rates (flow region), more intermolecular contacts are destroyed than are restored, and therefore there is a decrease in the density of contacts and consequently a decrease in the apparent viscosity. The initial dilatation zone and high apparent viscosity in the case of the Amon cultivar could be explained by the high intermolecular interaction between biopolymers. It has been reported that homogenization under high pressure leads to a significant decrease in the apparent viscosity and shear thinning of the 1% aqueous FSM solutions with increasing treatment pressure and time [62,77]. Such homogenization evidently disrupts the intermolecular network of FSM. Therefore, without taking into account the lowest shear rates (~0.1–0.3 s^−1^) corresponding to elastic deformations up to the yield stress (*τ < τ_Y_*), in the flow region (*τ > τ_Y_*), all FSM solutions exhibited pseudoplastic (shear-thinning) properties that were in agreement with previous reports on FSM solutions [14,16,17,18,19,75]. In the case of the Libra Bio cultivar, this deviation from Newtonian behavior was much less pronounced than in the other cultivars.

Polysaccharides are viscoelastic hydrocolloidal materials with properties of both solid and liquid bodies. In this context, the storage and loss moduli (*G*′, *G*″) have been used to represent the elastic and viscous properties of FSM, respectively [62,73]. The dependences of the values of *G*′ and *G*″ on angular frequency (*ω*) at 25 °C for 1% FSG solutions are represented in Figure 3a–c. It can be seen that for all samples, the values of both moduli were frequency-dependent over the entire experimental frequency range and increased with increasing *ω*, with *G*″ prevailing over *G*′. However, *G*′ increased more quickly than *G*″, and as a result, crossover of the *G*′ and *G*″ curves occurred at *ω*_c_ in the higher *ω* region. The samples of Libra Bio cultivar **lb17** and **lb18** exhibited the lowest values of *ω*_c_ (~17 and ~24 rad s^−1^) in comparison with the other FSM samples (~35–52 rad s^−1^), confirming the prevalence of the elastic component (Table 5). In contrast, sample **a15**, which showed the highest *τ_Y_* (see above), also demonstrated the highest *ω*_c_ (~52 rad s^−1^); this means that here a high apparent viscosity coexists with a smaller contribution of the elastic component. The strength of the intermolecular network formed by FSM polysaccharides is a prerequisite for viscoelastic properties in aqueous systems. It has been shown [77] that homogenization under high pressure leads to a sharp decrease in the moduli *G*′ and *G*″ for 1% aqueous solutions, in comparison with the native sample, mainly due to the destruction of this network.

Differences in shear-thinning behavior and the viscoelastic properties of FSM samples were evaluated using power functions (Appendix A). The rheological parameters obtained for the 1% aqueous solutions of FSM are shown in Table 5. As can be seen from the determination coefficients (*R*^2^ > 0.999), the power function showed good agreement for the downward flow profiles, as well as for the dependence of the loss modulus on the angular frequency (*R*^2^ > 0.997). The consistency index *K*_bw_ is related to the apparent viscosity and was maximal for **rb15** (~150 mPa·s^n^), followed by **a15** and **a16** (~107–109 mPa·s^n^), and minimal for Libra Bio (~12–15 mPa·s^n^); for the others it was in the range of ~48–65 mPa·s^n^. Conversely, the flow behavior index *n*_dw_ was maximal for Libra Bio (~0.92–0.95), minimal for rb15 (0.74), and in the range of 0.83–0.87 for the others. As previously found [62,79], homogenization under high pressure reduced *K*_dw_ and increased *n*_bw_ values due to the weakening of intermolecular associations typical of native FSM.

For all mucilages, the *K*′ values (~0.5–21.6 mPa·s^n^) were significantly lower than the corresponding *K*″ values (~9.8–138.6 mPa·s^n^). Furthermore, the *K*′ and *K*″ values were the highest for **rb15** (~21.6 and ~138.6 mPa·s^n^). Thus, among all mucilages, this sample had the strongest dynamic viscoelastic properties. In contrast, the Libra Bio samples **lb17** and **lb18** showed the lowest values of the *K*′ (~0.6–0.8 Pa·s^n^) and *K*″ (~10–16 mPa·s^n^) moduli. Therefore, the elastic properties were very weak, and the observed flow behavior was close to that of a Newtonian fluid, indicating the limited potential of Libra Bio as a thickener in colloid systems. It may be concluded that the FSM of this cultivar demonstrated weak steady-state viscous and weak dynamic viscoelastic properties. For the FSM samples, the values of the exponents *n*′ (~1.16–1.95) and *n*″ (~0.65–1.02) were characteristic of the typical behavior of a viscoelastic fluid. This means that physical cross-linking or entanglement of macromolecules occurred in these solutions. The exponent values for individual samples varied significantly and depended on the ratio of hydrocolloids, namely, polysaccharides and proteins. In addition, among the FSM solutions, the Libra Bio samples demonstrated the highest values of the exponents *n*′ (~1.84–1.95) and *n*″ (~0.89–1.02), corresponding to high frequency sensitivity [62].

The rheological tests used in this work are a powerful tool for the characterization of hydrocolloids, and they showed the specificity of individual flax cultivars, which is important for the use of FSM in various types of industries. The rheological parameters obtained for the FSM samples were partially different from those published previously [62]. Despite the high variability among the cultivars, the values that described the apparent viscosity of the FSM solutions were similar to those reported earlier. In contrast, the viscoelastic parameters were very different. They did not correspond to a weak gel but rather to a viscoelastic system with a crossover. These differences can be explained by the relationships of various factors, including the specificity of the cultivars used, storage time, the composition of hydrocolloids, and the impact of freeze drying.

### 3.4. Statistical Evaluation of Analytical Data

Statistical evaluation of the experimental data was performed using pair correlations and multidimensional discrimination methods to establish a relationship between the rheological properties and composition of FSM samples. These methods are suitable for such analytical tasks.

The matrix of Pearson correlation coefficients $r_{xy}$ for the 29 variables characterizing the composition and rheological properties of FSM samples is presented in Figure 4. A strong positive correlation was found between the molar ratios of Gal, Fuc, and Rha; all of them are units of RG-I. The molar ratios of these sugars demonstrated a marked negative correlation with NS, Ara, and Xyl; the last two showed a strong positive correlation as parts of AX. According to these correlations, Gal is associated with RG-I rather than with AX, as was mentioned earlier [11,12,13]. In contrast, Glc showed a weak negative correlation with all these sugars, therefore this is not a structural part of either AX or RG-I. The UA content demonstrated a weak positive correlation with Fuc, Gal, and Rha but was negatively correlated with Ara, Xyl, and Glc. Despite the fact that residues of glucuronic acid could be attached to AX [11,12,13], in the current study the correlations confirmed the association of UA as galacturonic acid with pectins (HG, RG-I) rather than with AX. Proteins showed a high positive correlation with %N, %S, and %K, a high negative correlation with %Na, %Mg, and %Ca, and some correlations with sugar composition.

The rheological characteristics summarized in Table 5 showed some correlations with the chemical composition of FSM. The power model parameters *K*_bw_, *K*′, and *K*″ showed a positive correlation with neutral and AX sugars and a negative correlation with RG-I sugars and proteins; the opposite is true for the corresponding exponents *n*_bw_, *n*′, and *n*″. The correlations of parameter *A* were insufficient. Therefore, AX supports shear thinning and the viscoelastic properties of FSM, while RG-1 and proteins may have the opposite effect. It was noted earlier that the neutral fraction of FSM contributed to a high apparent viscosity, as previously stated [19]. The yield stress *τ_Y_* and the thixotropic parameters *A*_hyst_ and *I*_hyst_ showed moderate positive correlations with Gal and UA and negative correlations with proteins. Therefore, the acidic component of FSM could be responsible for the yield stress and thixotropy. The crossover frequency *ω*_c_ showed a negative correlation with proteins and RG-I sugars, and a positive correlation with NS, Ara, and Xyl. These dependencies showed that individual macromolecular components can affect the ratio of the elastic and viscous properties of FSM in different ways: AX supports viscous features, while RG-I and proteins increase elasticity. The mentioned rheological parameters also showed correlations with inorganic elements, and therefore minerals can influence the rheological behavior of FSM solutions.

The multivariate statistical methods HCA and PCA were used to discriminate between FSM samples isolated from the flaxseed of different cultivars. The obtained results are illustrated in Figure 5a–d. According to the HCA dendrogram of similarity obtained by the Ward method of clustering algorithm (Figure 5a), the flaxseed cultivars were split into the three main clusters of Amon, Raciol/Recital, and Libra Bio cultivars. The dendrogram of similarity obtained for the variables (Figure 5b) demonstrates that the variables connected with proteins AX and RG-I are associated in tight clusters, which are significantly distant from each other. The rheological parameters of FSM solutions showed similarity to the specific chemical components of this substance. However, most of these parameters (*A*, *K*_bw_, *K*′, *K*″, *τ_Y_*, *A*_hyst_, *I*_hyst_) are associated with ash content, and therefore can be significantly affected by minerals. The relationship of these parameters with AX sugars is less pronounced. In contrast, *ω*_c_ is more closely associated with AX sugars and divalent cations, while the exponents *n*_bw_, *n*′, and *n*″ are tightly connected with proteins and, to a lesser extent, with RG-I sugars and UA.

The PCA score plot for the cases is shown in Figure 5c. The first two principal components PC1 (36.92% of total variants) and PC2 (25.76%) were found to be the most informative. The clusters of Amon, Raciol/Recital, and Libra Bio are well separated. PC1 divided Libra Bio samples (positive values) from the other flaxseed cultivars (negative or zero values), and PC2 separated Amon (positive values) from Raciol/Recital (negative values). The projection variables on the factor space are shown in Figure 5d. The variables showed a distribution in the factor plane similar to the corresponding Pearson correlation coefficients in Figure 5. For example, Ara and Xyl clustered together, demonstrating the presence of AX, while Rha, Fuc, and Gal were located together as components of RG-I and UA, but shifted slightly from them. The position of Glc was roughly perpendicular to the positions of the other neutral sugars, and proteins were found to be closer to RG-I sugars. The rheological parameters of FSM showed diverse distributions in this graph: (i) *τ_Y_*, *A*_hyst_, and *I*_hyst_ were found in proximity to UA and ash, (ii) *K*_bw_, *A*, *K*′, *K*″, and *ω*_c_ were located closer to the divalent cations and AX sugars, and (iii) *n*_bw_, *n*′, and *n*″ were located on the opposite side of *K*_bw_, *A*, *K*′, and *K*″, closer to proteins. Thus, the ratio between macromolecular components, namely, AX, HG, RG-I, and proteins, as well as the associated minerals, strongly affects the rheological behavior of FSM solutions. To explain this arrangement of variables, it can be assumed that acidic pectic polysaccharides, mainly HG and to a lesser extent RG-I, may be responsible for the yield strength and thixotropic properties of FSM, while neutral AX mainly determines the apparent viscosity and, to a greater extent, the dynamic viscoelasticity of this product. Proteins, on the other hand, have the opposite effect on these properties, but can aid in retaining the elastic component of the FSM in solution, possibly due to ionic interactions with polysaccharides, with the formation of non-covalent cross-links. As a larger highly branched macromolecular component, AX was considered to be responsible for shear thinning of FSM [5], although the purified AX fraction showed a much lower apparent viscosity than that of native FSM, and this is due to inter-connection with the acidic fraction. Therefore, not only the composition but also the intermolecular interactions in the native state can play a decisive role in understanding the rheological behavior of FSM in solutions, and the subsequent steps of homogenization, heating, drying, precipitation, or purification of the mucilage can significantly weaken or change these interactions [62].

## 4. Conclusions

The results of the present study suggest that the composition and rheological properties of flaxseed mucilage vary between different cultivars and harvest years. The rheological parameters that characterize the shear-thinning, thixotropic and dynamic viscoelastic behavior of this material in aqueous solutions demonstrated correlations with monosaccharide composition and the contribution of minor components such as proteins and minerals. It was found that the yield stress and the hysteresis loop were connected with the acidic polysaccharide fraction, which may include homogalacturonan and rhamnogalacturonan I. In contrast, the shear-thinning and especially the dynamic viscoelastic properties of flaxseed mucilage were determined mainly by the contribution of branched arabinoxylan, the highest molecular weight component. In addition, proteins also influence the viscoelastic properties and can retain the elastic component of the FSM in solution. Based on the fact that the composition and structure of both polysaccharide fractions, as well as the contribution of proteins, have a great influence on the rheological properties of flaxseed mucilage, the above-mentioned characteristics should be taken into account when considering effective applications for this material in future, both in the food industry and in cosmetics or medicine. For example, flaxseed mucilage with a high apparent viscosity, such as that of the Amon cultivar used in the current study, could be further investigated as a thickener for foodstuffs and cosmetics (lotions and creams). In contrast, mucilages showing a pronounced elasticity, such as those obtained from the seeds of the Libra Bio cultivar, could be further tested as fruit coatings or edible films. The rheological properties of flaxseed mucilages in this study were comparable to those of hydrocolloids such as guar gum or xanthan gum, which are used as thickeners. These are also highly branched polysaccharides but with different monosaccharide compositions, with the structures of galactomannan and mannoglucuronoglucan, respectively. However, aqueous solutions of flaxseed mucilage at a concentration of 1% did not form a gel, so this hydrocolloid cannot be proposed as a gelling agent similar to native starch, pectins, alginates, or sulfated galactans (agars, carrageenans). The present study provides useful information on the cultivar-dependent composition and rheological properties of flaxseed mucilage, which is important for its effective use as a food, cosmetic, or pharmaceutical additive. Based on the properties discussed, it is possible to determine the best grade of flax for a particular industrial application, for example, as an egg white or saliva substitute, since flaxseed mucilage exhibits characteristics reminiscent of those of these natural substances.

## Figures and Tables

**Figure 1 polymers-14-02040-f001:**
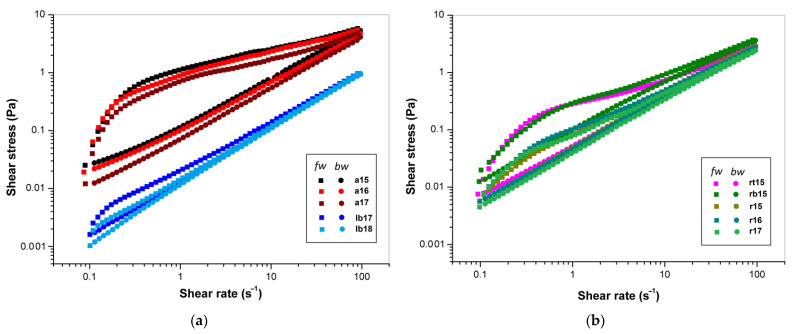
Plot of shear stress versus shear rate (on a double logarithmic scale) for 1% aqueous FSM solutions in the forward (squares) and backward (circles) directions: (**a**) Amon and Libra Bio cultivars; (**b**) Recital and Raciol cultivars.

**Figure 2 polymers-14-02040-f002:**
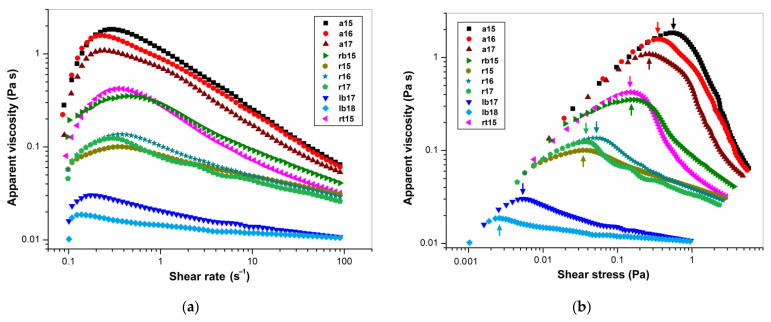
Steady shear flow rheograms (on a double logarithmic scale) for 1% aqueous FSM solutions: (**a**) plot of apparent viscosity versus shear rate in the forward direction; (**b**) plot of apparent viscosity versus shear stress in the forward direction. Arrows show yield points.

**Figure 3 polymers-14-02040-f003:**
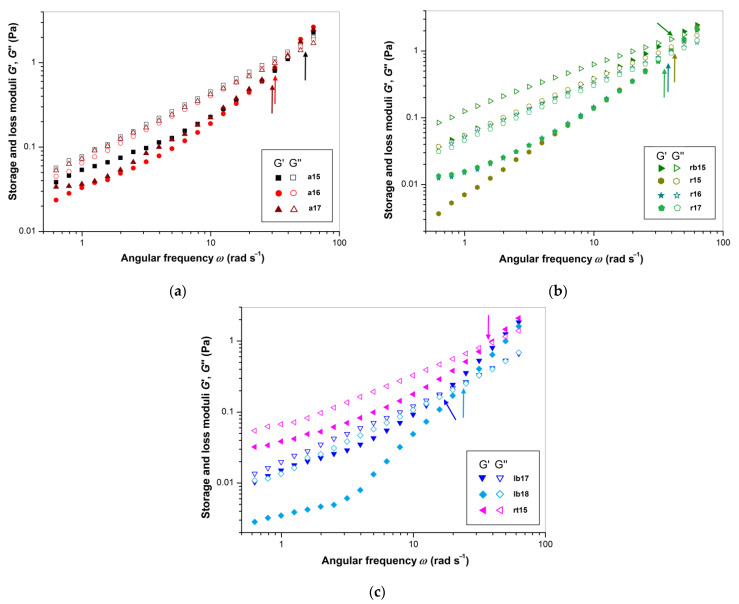
Dependences of the storage and loss moduli *G*′ and *G*″ on the angular frequency *ω* (on a double logarithmic scale) for the FSM samples in 1% aqueous solutions: (**a**) the Amon cultivar; (**b**) the Recital and Raciol cultivars; (**c**) the Libra Bio cultivar. The arrows point to the angular frequency *ω*_c_ at the crossover of *G*′ and *G*″ curves (*G*′ = *G*″).

**Figure 4 polymers-14-02040-f004:**
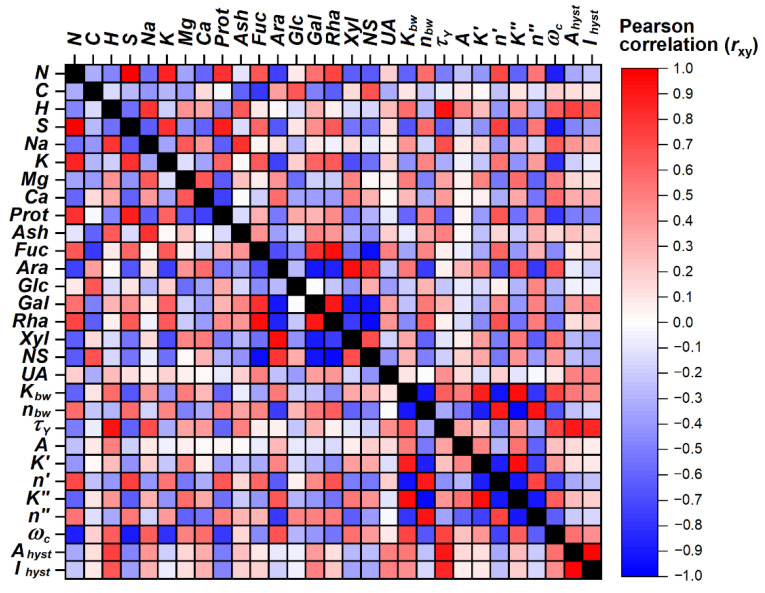
Correlation matrix of Pearson correlation coefficients $r_{xy}$ calculated for the analytical values obtained for the FSM samples (*N* = 10).

**Figure 5 polymers-14-02040-f005:**
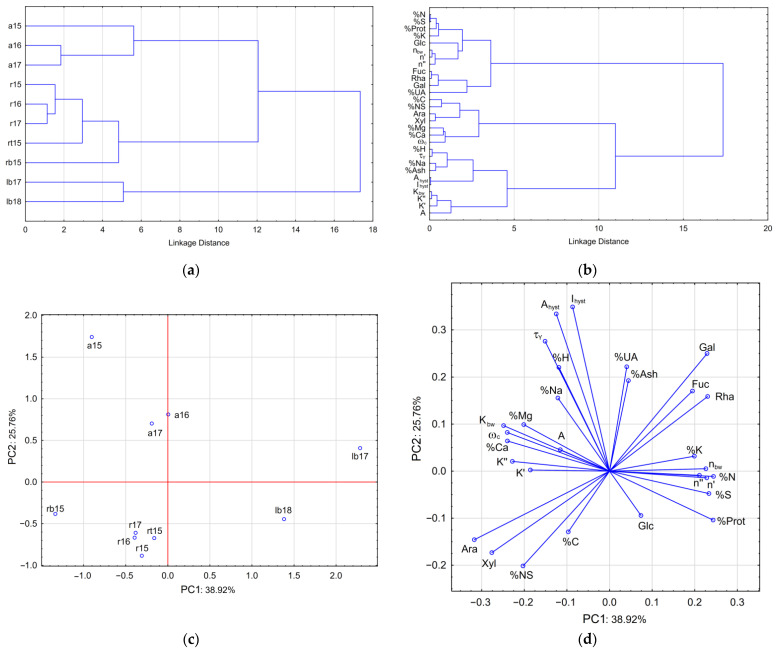
PCA projections of the cases (**a**) and variables (**b**) on the factor plane PC1 versus PC2. Dendrograms of similarity for flaxseed cultivars (**c**) and analytical values (**d**).

**Table 1 polymers-14-02040-t001:** Specification of flaxseed samples [56,57].

Sample	Cultivar	Breeder	Harvesting Year	Seed Color	Fatty Acid Composition
**a15**	Amon	Agritec (Šumperk, CR)	2015	yellow	low linolenic, high linoleic
**a16**	Amon		2016		
**a17**	Amon		2017		
**r15**	Raciol		2015		medium linolenic/ linoleic
**r16**	Raciol		2016		
**r17**	Raciol		2017		
**rb15**	Raciol Bio		2015		
**lb17**	Libra Bio	Limagrain Advanta Nederland, B.V. (Rilland, Holland)	2017	brown	high linolenic, low linoleic
**lb18**	Libra Bio		2018		
**rt15**	Recital	Laboulet Semences (Airaines, France)	2015		

**Table 2 polymers-14-02040-t002:** Extraction yield, pH, and electrolytic conductivity (κ) of FSM extracts from whole seeds of different flax cultivars (pH, κ: mean ± SD, *N* = 3).

Sample	Yield (% *w*/*w*)	pH	κ (mS cm^−1^) *
**a15**	7.9	5.79 ± 0.07 ^b^	1.645 ± 0.005 ^b^
**a16**	10.0	6.11 ± 0.01 ^d^	2.138 ± 0.004 ^f^
**a17**	7.5	5.98 ± 0.04 ^bcd^	1.712 ± 0.003 ^cd^
**r15**	8.6	5.84 ± 0.02 ^bcd^	1.724 ± 0.001 ^cd^
**r16**	8.7	6.07 ± 0.08 ^cd^	1.699 ± 0.006 ^c^
**r17**	7.0	5.86 ± 0.05 ^bcd^	1.959 ± 0.002 ^e^
**rb15**	9.1	5.83 ± 0.08 ^bc^	1.714 ± 0.001 ^cd^
**lb17**	6.4	5.47 ± 0.01 ^a^	1.741 ± 0.005 ^d^
**lb18**	9.7	5.48 ± 0.01 ^a^	1.694 ± 0.002 ^c^
**rt15**	10.0	5.37 ± 0.04 ^a^	1.395 ± 0.001^a^

* Different letters indicate significant differences between the FSM samples (*p* < 0.05).

**Table 3 polymers-14-02040-t003:** Elemental composition, protein, and ash contents of FSM from various cultivars (proteins, ash: mean ± SD, *N* = 3).

Sample	Content (% *w/w*)									
	Organic Elements				Mineral Elements				Proteins *	Ash *
	N	C	H	S	Na	K	Mg	Ca		
**a15**	**0.96**	37.18	6.13	0.12	1.52	2.80	0.47	0.77	4.83 ± 0.31 ^a^	19.63 ± 0.39 ^d^
**a16**	1.82	38.20	5.91	0.20	0.33	4.06	0.25	0.53	10.14 ± 0.16 ^d^	10.32 ± 0.22 ^a^
**a17**	1.42	37.93	5.87	0.16	0.41	3.62	0.33	0.84	5.88 ± 0.12 ^ab^	10.48 ± 0.21 ^a^
**r15**	1.42	37.78	5.71	0.18	0.38	1.96	0.27	0.52	9.88 ± 0.20 ^d^	10.12 ± 0.21 ^a^
**r16**	1.77	37.94	5.82	0.21	0.54	4.22	0.46	0.83	10.38 ± 0.21 ^d^	10.73 ± 0.22 ^a^
**r17**	1.40	37.71	5.74	0.18	0.40	3.45	0.35	0.77	6.65 ± 0.13 ^b^	11.09 ± 0.20 ^a^
**rb15**	1.28	37.68	5.88	0.16	0.58	3.64	0.43	0.59	7.98 ± 0.15 ^c^	10.14 ± 0.21 ^a^
**lb17**	3.04	37.04	5.76	0.33	0.13	5.86	0.27	0.40	16.95 ± 0.34 ^f^	12.93 ± 0.26 ^b^
**lb18**	1.91	37.77	5.80	0.22	0.60	4.47	0.21	0.43	12.60 ± 0.27 ^e^	15.22 ± 0.30 ^c^
**rt15**	1.39	38.31	5.87	0.19	0.21	3.12	0.17	0.60	13.55 ± 0.23 ^e^	10.97 ± 0.22 ^a^

* Different letters indicate significant differences between the FSM samples (*p* < 0.05).

**Table 4 polymers-14-02040-t004:** Monosaccharide composition of FSM extracted from various cultivars (NS, UA: mean ± SD, *N* = 3).

Sample	Molar Ratio (mol%) *							Content (% *w*/*w*)	
	Fuc	Ara	Man	Glc	Gal	Rha	Xyl	NS ^†^	UA ^†^
**a15**	4.75	13.27	0.15	16.00	25.28	8.87	31.69	29.61 ± 0.27 ^b^	21.07 ± 0.34 ^e^
**a16**	3.66	11.47	0.48	26.21	24.80	7.89	25.48	32.54 ± 0.25 ^c^	23.85 ± 0.42 ^f^
**a17**	4.36	12.56	0.35	20.17	25.25	8.76	28.55	30.57 ± 0.26 ^b^	13.37 ± 0.29 ^d^
**r15**	3.27	15.36	0.17	18.85	17.80	6.82	37.74	35.06 ± 0.29 ^d^	11.05 ± 0.23 ^bc^
**r16**	3.62	15.22	0	21.86	17.61	6.45	35.25	35.81 ± 0.32 ^d^	9.51 ± 0.21 ^b^
**r17**	3.71	15.17	0.18	18.98	16.89	6.43	38.65	37.88 ± 0.19 ^e^	21.96 ± 0.46 ^e^
**rb15**	3.31	15.71	0.29	20.37	18.72	5.93	35.67	37.49 ± 0.31 ^e^	10.25 ± 0.21 ^b^
**lb17**	6.39	8.26	2.14	17.32	30.00	12.56	23.31	22.81 ± 0.21 ^a^	21.77 ± 0.48 ^e^
**lb18**	3.78	9.17	0	29.72	24.98	8.45	23.90	33.25 ± 0.28 ^c^	6.88 ± 0.15 ^a^
**rt15**	3.20	14.82	0.18	24.65	18.56	6.87	31.73	38.16 ± 0.32 ^e^	12.62 ± 0.26 ^cd^

* Fuc: fucose; Ara: arabinose; Man: mannose; Glc: glucose; Gal: galactose; Rha: rhamnose; NS: neutral sugars; UA: uronic acids. ^†^ Different letters indicate significant differences between the FSM samples (*p* < 0.05).

**Table 5 polymers-14-02040-t005:** Rheological parameters measured for FSM samples from various flax cultivars (mean ± SD, *N* = 3).

Parameter	Sample									
	a15	a16	a17	r15	r16	r17	rb15	lb17	lb18	rt15
*K*_bw_ (mPa·s^n^)	109.02 ± 4.93 ^d^	106.67 ± 3.03 ^d^	68.62 ± 2.65 ^c^	64.76 ± 0.55 ^c^	56.21 ± 0.56 ^bc^	47.83 ± 1.61 ^b^	150.9 ± 3.84 ^e^	15.33 ± 0.23 ^a^	12.17 ± 0.33 ^a^	58.11 ± 2.47 ^bc^
*n* _bw_	0.838 ± 0.012 ^bc^	0.826 ± 0.007 ^b^	0.865 ± 0.003 ^c^	0.837 ± 0.002 ^bc^	0.844 ± 0.001 ^bc^	0.856 ± 0.005 ^bc^	0.738 ± 0.003 ^a^	0.921 ± 0.005 ^d^	0.954 ± 0.004 ^e^	0.848 ± 0.005 ^bc^
*R* ^2^	0.9998	0.9991	0.9998	0.9993	0.9997	0.9992	0.9995	0.9999	0.9996	0.9994
*τ*_Y_ (mPa)	571.3 ± 28.9 ^e^	348.7 ± 16.5 ^d^	268 ± 18 ^c^	34.7 ± 2 ^a^	52.3 ± 4.2 ^a^	38.1 ± 2.4 ^a^	153.1 ± 9.7 ^b^	4.7 ± 0.2 ^a^	2.9 ± 0.1 ^a^	146 ± 8 ^b^
*A* (mPa)	30.81 ± 0.80 ^abc^	209.0 ± 15.4 ^abc^	115.6 ± 9.8 ^ab^	1.467 ± 0.044 ^a^	12.20 ± 3.07 ^ab^	14.04 ± 2.21 ^ab^	43.26 ± 8.25 ^bc^	20.96 ± 1.94 ^abc^	2.41 ± 1.40 ^a^	51.03 ± 7.06 ^d^
*K*′ (mPa·s^n^)	6.27 ± 0.08 ^ef^	7.39 ± 0. 62 ^f^	4.12 ± 0.23 ^cd^	5.29 ± 0.10 ^de^	4.19 ± 0.10 ^cd^	3.28 ± 0.11 ^bc^	21.57 ± 0.26 ^g^	0.88 ± 0.03 ^a^	0.50 ± 0.02 ^a^	2.72 ± 0.09 ^b^
*n*′	1.411 ± 0.010 ^b^	1.412 ± 0.053 ^b^	1.515 ± 0.032 ^bc^	1.454 ± 0.031 ^bc^	1.487 ± 0.013 ^bc^	1.578 ± 0.051 ^c^	1.162 ± 0.006 ^a^	1.842 ± 0.005 ^d^	1.946 ± 0.002 ^d^	1.598 ± 0.011 ^c^
*R* ^2^	0.9966	0.9973	0.9870	0.9987	0.9986	0.9988	0.9986	0.9966	0.9976	0.9989
*K*″ (mPa·s^n^)	76.11 ± 1.41 ^d^	59.63 ± 0.86 ^c^	61.77 ± 1.58 ^c^	54.66 ± 1.26 ^bc^	56.79 ± 1.07 ^c^	47.82 ± 0.09 ^b^	138.6 ± 2.17 ^e^	16.04 ± 0.92 ^a^	9.80 ± 0.54 ^a^	55.22 ± 3.14 ^bc^
*n*″	0.771 ± 0.016 ^b^	0.831 ± 0.004 ^bc^	0.795 ± 0.007 ^bc^	0.828 ± 0.008 ^bc^	0.765 ± 0.005 ^b^	0.811 ± 0.053 ^bc^	0.651 ± 0.004 ^a^	0.892 ± 0.015 ^c^	1.020 ± 0.015 ^d^	0.778 ± 0.011 ^b^
*R* ^2^	0.9990	0.9998	0.9947	0.9992	0.9996	0.9983	0.9995	0.9975	0.9984	0.9995
*ω*_c_ (Rad·s^−1^)	52.29 ± 1.16 ^c^	39.75 ± 0.59 ^b^	37.74 ± 1.44 ^b^	41.99 ± 2.25 ^b^	37.32 ± 1.68 ^b^	35.24 ± 1.84 ^b^	39.16 ± 0.74 ^b^	17.47 ± 0.78 ^a^	24.32 ± 0.77 ^a^	37.17 ± 2.07 ^b^
*A*_hyst_ (Pa·s^−1^)	120.85 ± 6.63 ^cd^	136.70 ± 6.25 ^d^	107.79 ± 5.72 ^d^	10.13 ± 0.42 ^abc^	11.18 ± 0.30 ^abc^	10.32 ± 0.69 ^abc^	23.46 ± 0.12 ^bc^	5.30 ± 0.04 ^ab^	3.28 ± 0.01 ^a^	28.18 ± 0.49 ^a^
*I*_hyst_ (%)	29.80 ± 1.44 ^b^	36.14 ± 1.36 ^c^	34.69 ± 0.97 ^c^	6.45 ± 0.31 ^a^	7.45 ± 0.26 ^a^	7.66 ± 0.54 ^a^	10.43 ± 0.10 ^a^	10.10 ± 0.66 ^a^	6.51 ± 0.16 ^a^	16.16 ± 0.25 ^a^

Different letters indicate significant differences between the FSM samples (*p* < 0.05).

## Data Availability

The data presented in this study are available on request from the corresponding author.

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
