# Peer review of "Chemical Composition and Rheological Properties of Seed Mucilages of Various Yellow- and Brown-Seeded Flax (Linum usitatissimum L.) Cultivars"

_polymers, 2022, doi:10.3390/polym14102040_

Round 1

Reviewer 1 Report

Dear Authors,

Overall, it is a good manuscript, and the authors probably put a lot of work into its preparation. I like that you have added statistical analysis to the research. In my opinion, this manuscript can be published, but still requires a few corrections.

Detailed comments below:

Line 119: Add another utilitarian purpose to this research. Write down what your research will bring to, for example, industry? For example, will there be any new mucus-based products in the future ... etc.

Also justify why the aim of the work was to investigate 4 variants. My question is, why not 5 varieties, or, for example, 3 varieties? Explain it.

Line 123: I think you should also write what the linseed varieties were. Also add their Latin names.

Line 237: Only one exam has indicated the number of repetitions. So how many repetitions were there in the other studies?

Line 257: Write down how the normality of the distribution of empirical data was investigated. On this basis, you probably applied the Tucey test.

Line 684: In the part describing the test results, sometimes (in selected places) you should add (even one sentence) where your tests may be applicable. Also, whether your test results are near or far from the expected results. That's what's missing here. Moreover, the description is satisfactory.

Line 687: Add one more perspective. A conclusion that will show the prospects of your research for the future, etc.

Author Response

Overall, it is a good manuscript, and the authors probably put a lot of work into its preparation. I like that you have added statistical analysis to the research. In my opinion, this manuscript can be published, but still requires a few corrections.

Detailed comments below:

Line 119: Add another utilitarian purpose to this research. Write down what your research will bring to, for example, the industry? For example, will there be any new mucus-based products in the future ... etc.

Introduction: “Based on the results obtained, individual flax cultivars used in this work were considered as a raw material suitable for the production of FSM with certain rheological properties that could be prerequisites for specific applications as hydrocolloids for food and other industries.”

Also, justify why the aim of the work was to investigate 4 variants. My question is, why not 5 varieties, or, for example, 3 varieties? Explain it.

Linseed cultivars (variants) can be divided according to seed color (from brown to yellow) and fatty acid composition (ratio between linoleic and linolenic acids). Four variants used here represent brown-seeded (Libra Bio, Recital) and yellow-seeded forms (Amon, Raciol/Raciol Bio), and the same cultivars also represent three main groups according to fatty acid composition, i.e. low-linolenic (Amon), medium-linolenic (Raciol/Raciol bio) and high-linolenic (Libra bio, Recital) ones.

Line 123: I think you should also write what the linseed varieties were. Also, add their Latin names.

The linseed varieties used in this paper are summarized in Table 1. The binary Latin name (Linus usitatissimum L., Linaceae) mentioned in the title and Introduction represented the species (linseed, flaxseed) but not cultivars.

Line 237: Only one exam has indicated the number of repetitions. So how many repetitions were there in the other studies?

The number of repetitions was added to the tables representing statistical values.

Line 257: Write down how the normality of the distribution of empirical data was investigated. On this basis, you probably applied the Tukey test.

The normality of the distribution of empirical data was not tested because of the lack of samples and a large number of variables including parameters of power models. In contrast to microbial biomass, where the logarithmic component can have a certain weight, for various plant samples and products derived from them, it is usually assumed that the distribution of variables is close to normal. Therefore, analysis of variance and the Tukey test is often used to evaluate such data.

Line 684: In the part describing the test results, sometimes (in selected places) you should add (even one sentence) where your tests may be applicable. Also, whether your test results are near or far from the expected results. That's what's missing here. Moreover, the description is satisfactory.

The applicability of the tests and correspondence between obtained and expected values were added.

Line 687: Add one more perspective. A conclusion that will show the prospects of your research for the future, etc.

Conclusions: “The present study provides useful information on the cultivar-dependent composition and rheological properties of flaxseed mucilage, which are important for its effective use as food, cosmetic or pharmaceutical additives. Based on the properties mentioned, it is possible to determine the best grade of flax for a particular industrial application, for example as an egg white or saliva substitute, since flaxseed mucilage exhibits characteristics reminiscent of those of these natural substances.”

Reviewer 2 Report

The study is quite impressive and scientifically elaborated. I advise some minor corrections and it can be accepted after revising the following points:

- In the material and methods section, please add the description, especially the used concentration, of all protocols to make the essays more visible for lecturers.

- I suggest for authors to verify the parameters and conditions of the analysis and the machine information (name, manufacturers ...).

- Please, improve the quality of figures.

- Ensure the uniformity in the units (e.g. mg/mL, µL) throughout the MS.

- Correct some English mistakes before publication. Spelling should be revised thoroughly, many spelling mistakes detected.

- Check the references in accordance with the journal style.

Author Response

The study is quite impressive and scientifically elaborated. I advise some minor corrections and it can be accepted after revising the following points:

- In the material and methods section, please add the description, especially the used concentration, of all protocols to make the essays more visible for lecturers.

The detailed description of protocols was added.

- I suggest for authors to verify the parameters and conditions of the analysis and the machine information (name, manufacturers ...).

The parameters and conditions were added.

- Please, improve the quality of figures.

The quality of figures 5a-d was improved; the last figures are of high quality.

- Ensure the uniformity in the units (e.g. mg/mL, µL) throughout the MS.

The units were unified in the whole paper.

- Correct some English mistakes before publication. Spelling should be revised thoroughly, and many spelling mistakes detected.

The English grammar and style were revised and corrected

- Check the references in accordance with the journal style.

The references were corrected to be in accordance with the journal style

Round 2

Reviewer 2 Report

I can accept the paper in this form